



Exploring the effects of missing data on the estimation of fractal and multifractal parameters based
on bootstrap method
Xin Gao[1*], Xuan Wang[1]
[1]School   of   Urban-rural   Planning   and   Landscape   Architecture,   Xuchang   University,
Xuchang-461000, China
*Corresponding author : Xin Gao (gxin826@126.com)
**Keywords:** bootstrap, multifractal, interpolation method, time series, missing values
**HIGHLIGHTS**
•   Estimation   of   accuracy   for   the   parameters   of   fractal   and   multifractal   series   containing
15       missing values in the collection processes
•   Bootstrap statistical analysis of the fractal and multifractal parameters
•   A new resampling mechanism based on randomly gliding boxes
**Competing interests statement:** The authors declare that they have no competing financial
interests.










**ABSTRACT**
A time series collected in the nature is often incomplete or contains some missing values, and
statistical inference on the population or process with missing values, especially the population or
process having multifractal properties is easy to ignore. In this study, the simulation and actual
data were used to obtain the probability distributions of fractal parameters through a new bootstrap
resampling mechanism with the aim to statistically infer the estimation accuracy of the time series
containing missing values and four kinds of interpolated series. Firstly, the RMS errors results
showed that compared with the four interpolation methods for one parameter $H$ required for fBm
the direct use of the series with missing values has the highest estimation accuracy, while it shows
certain instability in the estimations of the multifractal parameters $C_1$ and $\alpha$, especially at higher
missing levels, however, the accuracy of the parameters estimated by preprocessing of piecewise
linear interpolation method can be improved; in addition, it is also concluded that $\alpha$ is more
sensitive to the changes caused by these processing than another parameter $C_1$. Secondly, the
effects on the ability of statistical inference for a population caused from the data losses are
explored through the estimation of confidence intervals and hypothesis testing by proposing a new
bootstrap resampling mechanism, and the conclusions showed that whether it is a mono-fractal
parameter or multifractal parameters, the large deviations from the estimates of original series
occur on the series with missing values when the losses are serious, while the defects can be
compensated by the preprocessing using PLI and PBI methods; similarly, although the results of
the incomplete series at the low missing levels are close to the original and PLI series, while at the
high missing levels, the probabilities of Type Ⅱ Errors of the neighboring values are unable to
ignore, but the PLI or PBI method can avoid the erroneous judgments.

**1 Introduction**

The scale invariance property has been known as a basic feature of natural phenomena,
which is always associated with the inherent properties of the physical world such as complexity,
non-smoothness, and irregularity. These phenomena with scale invariance are easily found in the
natural world, such as pulsation of turbulent flow in pipelines, accumulation of minerals in the
crust of the earth, volatility of financial market prices and winding coastline. As for the





universality of this law, as pointed out by Cheng (2008) in the evolutions of a metallogenic system,
various physical, chemical, and biological processes that take place in the earth system are
interrelated, influenced and restricted mutually, which constitute a self-organizing structure, so as
to make the system vary from equilibrium to far from balance, then to a critical state, and finally
until a new cycle. For these phenomena, the traditional linear theory lacks the physical basis, and
the process-based differential models are slightly less efficient, while fractal theory provides an
effective method for describing this feature. So far, a variety of fractal models have been proposed,
the most famous being the $\alpha$-$f(\alpha)$ model based on measure theory and the co-dimensional model
$\gamma$-$c(\gamma)$ based on probability theory (Evertsz and Mandelbrot, 1992; Schertzer and Lovejoy, 1987).
Observations of the natural world such as atmospheric environment, land cover and
meteorological factors are usually recorded as time or spatial series to study the evolution of the
earth system. As mentioned above, scale invariance is universal in the natural world, therefore the
data collected from these natural phenomena such as $PM_{2.5}$, surface temperature of the earth, and
DEM generally possess fractal properties. Statistical inference of fractal parameters such as
confidence interval estimation, hypothesis testing, etc, is important for accurate modeling. For
example, the Hurst index $H$ of Brownian motion is 1/2, while $0<H<1/2$ or $1/2<H<1$ indicate it is a
fractional Brownian motion with negative or positive long-range dependence, in addition, for a
generalized cascade series, when $\alpha=2$, it is a log-normal process, and when $1<\alpha<2$ or $0<\alpha=<1$, it
is a log-Levy process, while $\alpha=1$, then it is a Cauchy process. A lot of studies about the statistical
inferences of the parameters for time series with fractal or long-range dependence properties has
been done. For example, Wendt et al. (2007) used the bootstrap resampling method to
quantitatively study the estimated accuracy of multifractal parameters, and Gao et al. (2002)
studied the estimation of the spectral function for non-stationary Gaussian process with stationary
increments by constructing an estimator of asymptotic normality through the Gauss-Whittle
function; in addition, on the basis of the in-depth analysis of the causes of sensitivity and deviation
for calibrated spectral functions, a modified Jackknife estimator is tested to reduce the bias by
Gaume et al. (2007). However, due to human or mechanical factors such as equipment
maintenance, power failure, and improper human operations, the data obtained by users are often
incomplete and there will be a large number of missing values. The effects on the estimates of
parameters caused from the missing data for time series with fractal properties are easily ignored,



therefore, it is necessary to carry out statistical inference on fractal parameters for the time series
containing missing values.
Lack of data makes it difficult for a series of statistical methods to be used properly, because
the preconditions for their use are compromised. Many imputation methods have been developed
in the past to efficiently estimate a parameter of interest $\theta$ in a missing data situation, and to assess
the variability of the estimates $\hat{\theta}$, i.e., multiple imputation, Bayesian imputation or commonly
used interpolation methods. Considering singularity or irregularity is the essential feature of
fractal or multifractal data, while most interpolation methods are implemented by using some
linear or nonlinear models, and there is a problem, namely, it is whether the production of the
relative regular data will have the positive or opposite effect on the parameter estimation. For
example, researchers who are engaged in comparative politics or international relations, or others
with the incomplete data, have been unable to complete the data because the best available
imputation methods work poorly with the time series cross-section data structures common in
these fields (Honaker and King, 2010). However, people always make the statistical inference
after interpolating such data, and the common statistical inference methods cover parametric and
non parametric methods. Robins and Wang (2000) have developed an estimator of the asymptotic
variance of both single and multiple imputation estimators, especially for the variance estimator,
which is consistent even when the imputation and analysis models are misspecified and
incompatible with one another; considering the variance estimator proposed by Rubin, can be
biased when the imputation and analysis models are misspecified and/or incompatible, Hughes et
al. (2016) explored four common scenarios of misspecification and incompatibility through a full
mechanism bootstrapping method and modified Rubin's multiple imputation procedure.
Statistical inference involves making propositions about a population based on estimators
constructed from some samples of the population. Compared with parametric methods,
non-parametric methods require less assumptions about probability distributions made about a
population or process. However, they are computationally intensive, and lie in using computers to
resample a large number of new samples from one original sample, so as to obtain the estimates
based on the sample distributions. For fractal or multifractal series formed by infinite subdivisions,
the formulas of the parameter estimators for statistical inference are more complex and need to





make some assumptions, but non-parametric methods can avoid the derivations of the formulas.
Obviously, the accuracy of the non-parametric statistical inference depends on the degree of the
resemblance to the original sample for the resampling samples. The study by Wendt et al. (2007)
indicates that the parameter confidence intervals calculated by the bootstrap method well cover the
intervals simulated by Monte Carlo method, i.e. the resampling samples can well reflect the
characteristics of the population. When pursuing the accuracy by performing statistical inference,
except directly using the series with missing data for estimation, in order to reduce errors, it is
often necessary to perform interpolation or imputation preprocessing. Here we will adopt four
kinds of interpolation methods to deal with the missing data including piecewise linear
interpolation (PLI), piecewise cubic spline interpolation (PCSI), piecewise cubic Helmit
interpolation (PCHI) and piecewise Bessel interpolation (PBI). Based on the five kinds of series
generated from simulation and experimental data, the performances of statistical inference will be
studied by proposing a new resampling mechanism with the purpose to obtain the quantitative
study of the accuracy of the parameters for the series containing missing values with fractal or
multifractal properties.

**2 Methodology**

In order to test the statistical performances of the parameters for the time series with missing
values, here we will apply two kinds of simulation data with a priori known and controlled scaling
properties, fractional Brownian motion and generalized cascade process, the generation and
estimation of which are introduced as follows.
**2.1 Simulation and estimation of fBm and cascade process**

Fractional Brownian motion is a typical Gaussian process with self-similar property,
long-range dependence, stationary increments and regularity. The self-similar feature is
characterized by the parameter $H$, also called Hurst index, which is between [0, 1]. The larger the
$H$ value is, the smoother the process will be, on the contrary, the smaller the value is, the coarser
the process will be. Except the special case $H=1/2$ when the process is Brownian motion, if $H>1/2$,
the increments are positively correlated, and while $H<1/2$, the increments are negatively correlated.





There are many kinds of simulation methods for fractional Brownian motion, which can be
divided into two kinds: exact and approximate. The exact includes Hosking method (Hosking,
1981), Cholesky method and Davies and Harte method (McLeod and Hipel, 1978; Davies and
Harte, 1987), while the latter has stochastic integral method, summation method, random
displacement method and wavelet decomposition (Mandelbrot and Van Ness, 1968; Norros, et al.,
1999; Meyer et al., 1999). Here we will adopt the simulation method based on the wavelet
coefficients, which are synthesized from the decomposition terms of the Gaussian white noise
using wavelet transformation.
The estimation methods commonly used for $H$ are R/S analysis, regression residual variance,
wavelet estimation, spectral analysis and periodogram. The regression residual variance method is
widely used in the estimates of fractal features with the following specific steps: suppose there is a
series $x_t$, where $t$=1, 2...$N$, firstly, the cumulative difference is calculated: $Y_t = \sum_{t=1}^{n} x_t - <x>$ ;
Secondly, divide the series into subintervals with length $N_s$=int($N/s$) that do not overlap, and in
order to reduce the errors the series is flipped and repeated, so $2N_s$ subintervals are obtained,
where $s$ is the length of each segment; thirdly,each subinterval is subjected to the elimination of
the trend from the regression operation to yield the series: $Y_s(t)=Y(t)-p_v(i)$, where $p_v(i)$ is the
polynomial trend obtained by fitting $Y(t)$ using the least square method to the subinterval of the $v$th
segment; finally, the variance $F_s^2(v)=<Y_s^2(t)>$ is calculated for each interval, and we can get the
fluctuation function $F_s = [\frac{1}{N}\sum_{v=1}^{N_s} F_s^2(v)]^{1/2}$ in the whole interval. Then the parameter estimates can
be obtained through the relationship between the fluctuation functions and the scale $F(s) \propto s^h$. In
addition, $F^n(s)$ can be obtained by detrending using $n$-order polynomial fitting function, here $n$ is
set to 1.
From the view of the construction of a fractional Brownian motion, the process involves the
iterative sums of random quantities, and the corresponding is the iterative products of random
quantities with the following specific construction process shown in **Fig.1**: suppose there is a unit
quantity $I_0$, which scale is viewed as l, first split $I_0$ and its scale become l/2, yielding two values,
$I_0*\eta_{11}$ and $I_0*\eta_{21}$, and $\eta_{11}$ and $\eta_{21}$ are random variables followed a probability distribution with





$<\eta>=1$, where $<>$ means expectation; second, the analogy is continuously iterated by $k$ times and
the result $I_0 \prod_{i=1}^{k} \eta_{f(j,i),i}$ is obtained, where $j=1, 2, \ldots, b^k$ is the positional index of the layer $k$, $i$ is
the number of a layer, where at this level the scale is $b^{-k}$, and $f(j,i)$ represents the position in the $i$-th
layer, with the form $f(j,i)$=roundup$(j/b^{(k-i)})$ (Gaume, et al., 2007). The series generated from the
above process is called cascade process which is dominated by the multi-fractal behavior, and the
simulation data in this study is generated by the method provided by Schertzer and Lovejoy

(1987).


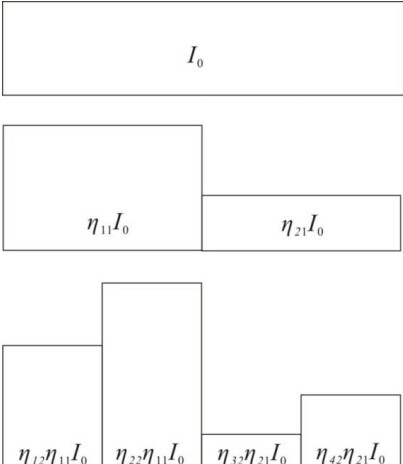


**Fig.1.** The generation of cascade process

Compared to mono-fractals, characterizing multi-fractals requires many parameters. Suppose
there is a parameter $\gamma$, there is $\Pr(x>\lambda^\gamma)\sim\lambda^{c(\gamma)}$, where $\gamma$ is a singular value, and $c(\gamma)$ is the
co-dimension function which is concave. Another widely used parameter is scaling function $K(q)$,
which is related to $c(\gamma)$ by Legendre transformation pairs. In order to simplify the multifractal
parameters, Schertzer and Lovejoy (1987) introduced an universal generalized multifractal model
and simplified the parameters into two parameters: $\alpha$ and $C_1$, where the former represents the
strength of multifractal, and the latter expresses sparse features. The accurate estimates of $C_1$ and
$\alpha$ are an important task for multifractal analysis, usually the methods of which have the moment
method and the double-trace moment method. Because involving lots of operations, the moment



estimation method is used in the following study with the process which is as follows: firstly
calculate the scaling function $K(q)$: supposing a series $x_t$, divide the series into different interval
lengths, $x_{s,n}$, where $s$ is the length of each segment, $n$=1, 2..., $Ns$, $Ns$=int(N/s), and next calculate
$x_{s,n}{}^q$ for different length $s$ respectively, then the $K(q)$ function can be estimated by the relation
$<x_{s,n}{}^q> \sim \lambda^{K(q)}$, where $\lambda$=length$(x_t)/s$, length$(x_t)$ is the length of the series; secondly, use the formula
$K(q) = \dfrac{C_1}{\alpha - 1}(q^\alpha - q)\,, 0 \le \alpha \le 2$   for the nonlinear fitting to get the parameter estimates.

**2.2 Interpolation methods**

The interpolation methods involved in this paper include piecewise linear interpolation (PLI),

piecewise cubic spline interpolation (PCSI), piecewise cubic Helmit interpolation (PCHI) and
piecewise Bessel interpolation (PBI). For PLI method, the missing values can be achieved by
constructing linear functions between the adjacent known points around them, which does not
consider the derivative values of the known points, and the accuracies of the interpolation points is
related to the distance between the known points. PCHI method not only requires that the values at
the two known points is equal to the values of the interpolation function, but also that the
derivative values are equal, thereby improving the smoothness of the interpolants. Unlike the
above interpolation methods, PCSI method requires that the second derivative be continuous at the
known points in each interval, meaning that the interpolant is smoother than the previous two
methods. Since there are many research on the above three kinds of interpolation methods, only
the calculation formulas of PBI method are given here.

Unlike above interpolation methods, with the fact that the derivative values at the known

points are not required to be equal to that of the interpolant for PBI method, its shape is controlled
through the control points, which must be on the tangent line to the fitted curve at the known
points. Let us take a look at the calculation process of PBI with $a \le x_1 < x < x_k \le b$, where $a$ and $b$ are
respectively the beginning and the end point of the series, $x_1$ and $x_k$ are the two known points, and
$x$ are the missing points. In addition to the two known points, it requires to know two control
points located between the known points. Denoting the two known points and the two control
points as $P_0$, $P_3$ and $P_1$ , $P_2$ respectively, there are $P_0$=$y_1$ and $P_3$=$y_k$, and $P_1$ and $P_2$ are related to the
tangent and can be written as functions of the derivatives at the known points, thus, there are

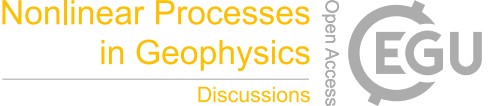



$$P_1 = P_0 + 1/3(x_k - x_1)f'(x_1), \quad P_2 = P_3 + 1/3(x_k - x_1)f'(x_k),$$ (1)
where $f'(x_1)$, $f'(x_k)$ can be obtained by the differences between the known points and their
neighboring points. After the four values at the four points are obtained, the Bezier curve can be
written as a basis function form $B(t) = (1-t)^3 P_0 + 3t(1-t)^2 P_1 + 3t^2(1-t)P_2 + t^3 P_3$, where
$t = (x - x_1)/(x_k - x_1)$.

**2.3 A resampling mechanism based on randomly gliding boxes**

The traditional statistical inference is based on the normal distribution, while multifractal
series usually have thick-tailed features, and the robust estimation can not be obtained by using the
normal distribution test. The parametric statistical inference is based on the theoretical probability
distribution of a population, that is, it is necessary to obtain the certain probability distribution in
advance, and from the generation process of the cascade we can get it is difficult to obtain the
probability distribution function of the cascade series. On the contrary, the non-parametric method
need not to know the probability distribution function of the statistic, and can perform hypothesis
testing on the condition that the actual distribution information of the series is poorly known, and
can guarantee the robustness of the estimation. The key to the non-parametric method is to obtain
the resampling distribution of the parameters from the original sample through resampling
methods, and then use the resampling distribution to perform statistical inference, where the
common resampling methods are bootstrap, jackknife, and Monte Carlo, etc..
In order to obtain the resampling distribution of an estimator, a new bootstrap resampling
mechanism is introduced (**Fig.2**). As mentioned above, the parameter estimation process involves
the series being divided into multiple non-overlapping intervals at different scales, and each
interval contains the same number of the values. Imagine that if the starting point of the
calculation for the series is different, and the estimation results obtained will be different. In
practice, the origin-based estimation were used in many studies, and this caused the waste of the
estimated information from original series. If a large number of random numbers are used as the
starting points, then multiple estimates of the parameters can be obtained to form the resampling
distribution. Obviously, this resampling method is achieved by the glide of a group of boxes
controlled by a series of random numbers, therefore we could call it randomly gliding boxes (RGB)

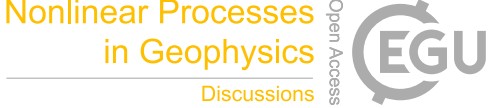



method. It is easy to see that the resampling mechanism can ensure that the resampling
distribution function is similar to the original distribution, of course, and this method will be
limited by the length of the series, the same idea can be seen in the Cheng's study (1999).

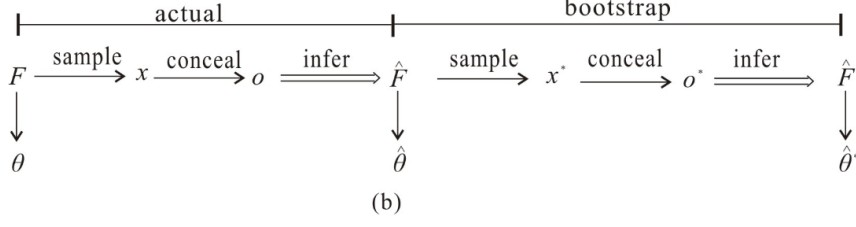


**Fig.2.** A new resampling mechanism realized by constantly determining the starting points controlled by a series of

random numbers, also called randomly gliding boxes method.


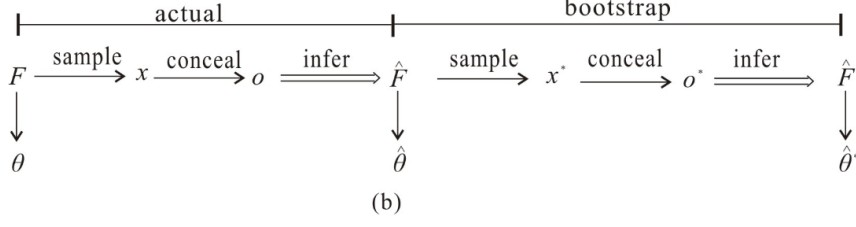


Fig.3. Two kinds of bootstrap logics by Efron (1994)


The main purpose of various sampling logics or mechanisms is to reduce the bias of an

estimator $\hat{\theta}^*$ of $\theta$, which depend on the adequacy of the use of population information,
concealment, and a sampling procedure, etc. As shown in **Fig.3(a)**, suppose there is a population $F$,
$F = \{X_j, j = 1, \cdots, N\}$, where $X_j$ denotes a random sample having one or several random variables,
then the population $G$ with some concealment in some members is generated through a
concealment process $o = c(x_i)$. $\theta_F = s(F)$ is the inference we need, where in our study $s(F)$ is a fractal




or multifractal parameter. A sample $o$ with size $n$ is obtained from the population $G$, and after
imputation or interpolation the empirical distribution $\hat{G}$ can be got from it, so we can get
$\hat{\theta} = t(\hat{G})$ as an estimate of $\theta_F$. The bootstrap inference begins with $\hat{G}$ instead of $G$, and repeat
the above procedure to get a bootstrap estimate $\hat{\theta}^* = t(\hat{G}^*)$ of $\hat{G}^*$ corresponding to each
replication $o^*$. This mechanism does not fully use the population information, merely based on the
establishment of the first sampling, that is, there is a small connection in the method with the
required knowledge of $\theta_F$ or $\theta$. The full bootstrap mechanism diagrammed in **Fig.3(b)** is more
directly than in Fig.3(a), in which a random sample $x$ is drawn and processed by a concealment
$o=c(x_i)$ to obtain the observed data $o$, and then the parameter $\theta=s(F)$ is estimated by $\hat{\theta} = s(\hat{F})$. To

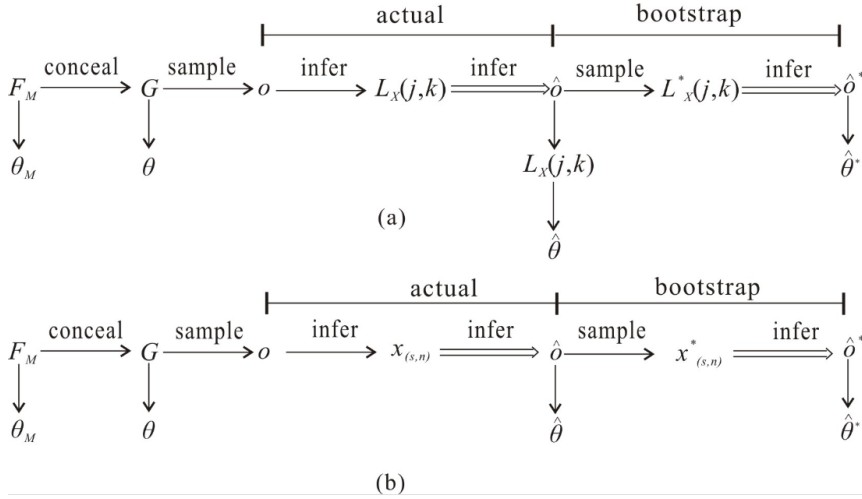


Fig.4. Our proposed bootstrap logic and another by Wendt

be different of this method is by repeating the whole process, i.e., sampling, concealment and
inference to yield bootstrap replications $\hat{\theta}^* = s(\hat{F}^*)$. In our study, for the fractal or multifractal
estimation, use the above simulation method to generate a sample set with a capacity of $N$ from
the known parameter values, denoted as $F_M$, and an concealment procedure is used to eliminate a
part of data for each sample to obtain the sample set $G$ containing missing values, which can be





considered as observations. Another four sample sets are generated when apply the four
interpolation methods to the sample set G, and therefore *N* estimates are available for each sample
set, which are also called Monte Carlo estimates. Before using our proposed bootstrap method,
given the uniqueness of the acquired time series one of the Monte Carlo samples is selected for
statistical inference. The key to our proposed method shown in **Fig.4(a)** is to obtain the probability
distributions of the estimated values by multi-segmenting the time series by random numbers,
which is different from the resampling of the moment estimators at different scales on the basis of
one-time segmentation shown in **Fig.4(b)**. The advantage is that it is easy to control the biases of
multifractal parameters, especially the time series with higher sparseness.

**2.4 Bootstrap confidence Intervals and hypothesis testing**

After obtaining the bootstrap resamlpling distribution, and then we can perform the interval

estimation and hypothesis testing on the estimators. Common bootstrap confidence interval
methods are normal approximation method, percentile method, bias-corrected percentile method,
and percentile-*t* method, and this study will use the percentile and percentile-*t* methods. Supposing
the function of bootstrap distribution of $\hat{\theta}$ is $F^*(\hat{\theta}^*)$, there is $P\{\theta^*_{a/2} < \hat{\theta} < \theta^*_{(1-a)/2}\} \geq 1-a$,
and the confidence interval of $\hat{\theta}$ is $[\theta^*_{a/2}, \theta^*_{(1-a)/2}]$, also known as the confidence interval with
a confidence level of 1-$\alpha$, where $\alpha$ is quantile. The meaning of the confidence interval is that if
there are multiple samples, the probability that the confidence interval calculated by each sample
contains the true value is 1-$\alpha$. The percentile confidence interval has two shortcomings, one is that
when the sample size is small, its performance is poor; the second is the need to make assumption
that the bootstrap distribution is an unbiased estimate in advance. In order to avoid the above
problems, the percentile-*t* method proposed is as follows: firstly transform $\hat{\theta}^*$ into a standard
variable $t^*$: $\hat{t}^* = (\hat{\theta}^* - \hat{\theta})/\hat{\sigma}^{**}$, and the bootstrap distribution of $t_B^*$ is determined through
resampling, where *R* is the number of resampling times; secondly, like the percentile method, we
need determine the statistical values of $t^*$ at $\alpha/2$ and 1-$\alpha/2$, and then combining with the
parametric test we can get *t*-test confidence interval $[\hat{\theta} - \hat{\sigma}^{**}\hat{t}^*_{B,1-\alpha}, \hat{\theta} - \hat{\sigma}^{**}\hat{t}^*_{B,\alpha}]$, where $\hat{\sigma}^{**}$ is
the standard deviation obtained by double sampling, that is, we need to take the second sampling
for *S* times after the first sampling, *S* is the number of times of the secondary sampling.



Another problem of statistical inference is hypothesis testing, for example, if $H$=1/2, the

series is a Brownian motion, while for $H \neq 1/2$ it is a fractional Brownian motion; for cascade

series, when $\alpha$=2, it follows log-normal distribution, otherwise it follows log-Levy distribution.

Hypothesis testing of a parameter is firstly to set a null hypothesis, $H_0$: $\theta = \theta_0$, and an alternative

hypothesis $H_1$:$\theta \neq \theta_0$, and then construct a hub statistic that contains the test parameter, where

percentile and percentile-$t$ statistics are used with the form $\hat{t}_B = \hat{\theta} - \theta_0$ and $\hat{t}_s = \dfrac{\hat{\theta} - \theta_0}{\hat{\sigma}*}$

respectively. When the condition $\Pr\{t \in T \big| P_t^{H_0}\} = 1 - \alpha$ is met, the original hypothesis is

accepted, otherwise the original hypothesis is rejected, where $T$ is the accepted domain, and $\alpha$ is a

quantile, which is called the significance level. However, since the decision is made from a sample,

when $H_0$ is actually true, it may be possible to make a decision that rejects $H_0$, this is the error

denoted by $\alpha_{error} = P_t\{x \in T \big| H_0\}$, which is also called Type Ⅰ Error; similarly, the error

$\beta_{error} == P_t\{x \in \overline{T} \big| H_1\}$ is called Type Ⅱ Error, where $\alpha_{error}$ and $\beta_{error}$ are the probabilities of Type

Ⅰ Error and Type Ⅱ Error respectively. If the theoretical probability distribution function of the

estimators is not available, there is no way to calculate the probability that the statistic falls within

the accepted domain. But the acceptance domain can be obtained through the bootstrap method,

thus the test statistic become $\hat{t}_B^* = (\hat{\theta}^* - \hat{\theta})$ and $\hat{t}_s = \dfrac{\hat{\theta} - \theta^*}{\hat{\sigma}^{**}}$, and the corresponding accepted

domains for the percentile and percentile-$t$ methods are $[\hat{t}_{B,\alpha}^*, \hat{t}_{B,1-\alpha}^*]$ and $[\hat{t}_{S,\alpha}^*, \hat{t}_{S,1-\alpha}^*]$.

**3 Results and discussions**

For the bootstrap estimation, the relevant parameters will be set as follows: $R$=500, $S$=300,

sample size $N$=1000, and the quantile $\alpha$ is set to 0.05, if it is a bilateral test, then the half is 0.025.

Firstly, the simulation data of fractional Brownian motion and generalized multifractal series with

the known values for parameters are generated based on above mentioned generation methods,

here we adopt $H$=0.6 for the fBm data and $\alpha$=1.6 and $C_1$=0.2 for the multifractal series, both of

them having 2048 data points (**Fig.5**). Since these series are not got any processing, so they are

labeled as the original series. Secondly, the series containing missing values are generated, which




will be controlled by the missing degree which is represented by five levels, and the higher the
level is, the more the missing values will be. The specific implementations are to randomly
remove $T$ data segments from the original series, where $T$ in turn takes the values of 30, 50, 70,
100, and 150 respectively, and the number of each data segment is controlled by a random number
ranging from 1 to 50. Thirdly, once the series with missing values are generated, then the four
types of interpolated series are generated by the four interpolation methods.

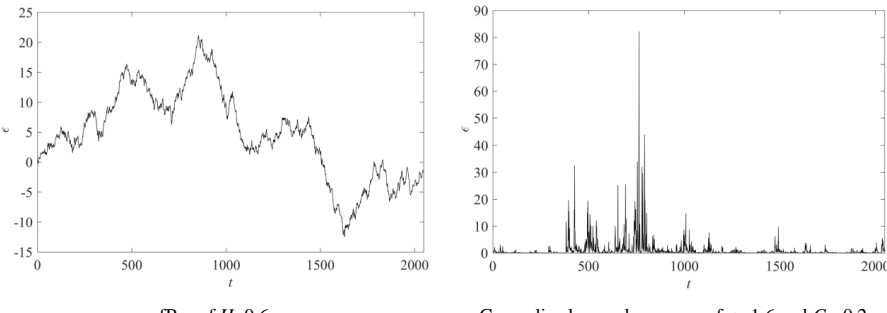

fBm of $H$=0.6                    Generalized cascade process of $\alpha$=1.6 and $C_1$=0.2

**Fig.5.** The sample data generated by fBm and cascade models


**3.1 Validation of mono-fractal data with missing values**

**Table 1** RMS errors of parameter $H$ calculated for the series containing missing values (SCMV) and the four kinds
of interpolated series at five missing levels

| Data Types | Level1 | Level2 | Level3 | Level4 | Level5 |
|---|---|---|---|---|---|
| SCMV | 0.001 | 0.0013 | 0.0006 | 0.0011 | 0.0009 |
| PLI | 0.0042 | 0.0071 | 0.0125 | 0.016 | 0.0194 |
| PCSI | 0.078 | 0.0923 | 0.114 | 0.1198 | 0.1371 |
| PCHI | 0.0468 | 0.0565 | 0.074 | 0.0784 | 0.0942 |
| PBI | 0.0007 | 0.0003 | 0.0052 | 0.0106 | 0.0144 |


For a biased estimation statistic, the root mean square (RMS) error is widely used to

quantitatively describing the estimation accuracy of the parameter. The RMS errors listed in **Table**
**1**, except the RMS error of the original series being 0.0028, calculated for the given five kinds of
series are compared to show the accuracy of the parameter $H$. It can be seen that the series
interpolated by PBI method and the direct use of the series containing missing values work best,
while for the series with fewer missing values, the accuracy of PBI method is even higher. The





next best performances are PLI and PCSI methods, while PCHI has the largest deviations. In
general, with the increasing number of the missing values in the sample series, the accuracy of the
estimates gradually decrease. From the estimates directly using the incomplete data, we can see
that the change of the original series caused by the interpolation methods may be one of the
reasons for the increase of the errors.

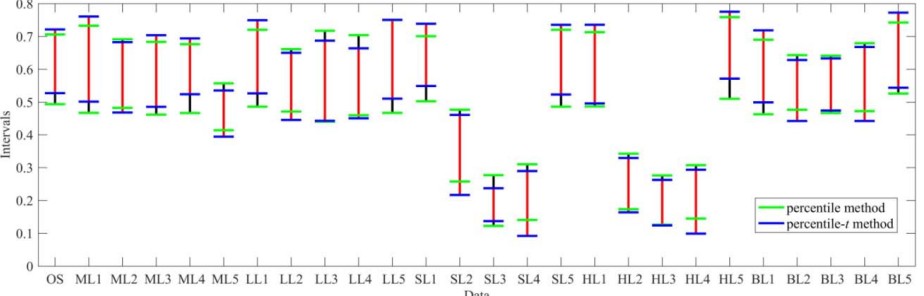


**Fig.6.** The percentile and percentile-*t* intervals of *H* estimated for the six kinds of time series at five levels based
on RGB resampling mechanism (OS denotes the selected sample from the original samples, ML denotes the series
containing missing values and the number 1 means the level, and LL, SL, HL and BL denote the series interpolated
by PLI, PCSL, PCHL and PBL methods respectively)

Because the sample series collected or obtained in the real world usually is unique, therefore
firstly we need choose one sample from the Monte Carlo simulations and then perform statistical
inference, and the percentile and percentile-*t* intervals estimated for various series using the RGB
method are shown in **Fig.6**. From the percentile or percentile-*t* perspective, it can be seen that as
the amounts of the missing data increase, the estimated values for the series with missing values
gradually decrease and constantly deviate from the original estimate, and visually the correct
estimate of *H* is not obtained by using directly the incomplete data when the amount of the
missing data increases to a certain extent. The endpoint estimates of the confidence intervals are
also random variables, based on the comparisons of the percentile estimates of the two endpoints
in each interval for the five kinds of series, we can see that the PLI and PBI series have the best
performances, especially at the high missing levels, which obviously compensate for the impacts
caused by the missing data, and the same conclusion can be obtained from the percentile-*t* method,
therefore this indicates that the estimation accuracy of incomplete data can be improved by the
preprocessing using PLI and PBI methods. In addition, it can be seen from the estimates of the




original series that the estimated intervals using percentile method shift to the left compared with
the *t*-method, which are experimentally by Monte Carlo method found to be caused by the
variances of the series. Moreover, from the estimated results of *t*-method, it can be also concluded
that the stability using PLI method for the right endpoint estimation is robust.




**Fig.7.** The coverage ratios of *H* estimated for the six kinds of time series at five levels by using percentile and
percentile-*t* methods, i.e., the probabilities of the estimates of six kinds of simulation data with N=1000 fall within
the intervals of the selected sample from original samples by RGB mechanism

The reliability of the parameter *H* estimated is also evaluated through the coverage of the
estimated intervals for the incomplete or processed samples, i.e., the probabilities of the Monte
Carlo estimates for the incomplete and interpolated time series falling within the bootstrap
intervals of the selected sample extracted from the original samples (**Fig.7**). The results in the
figure clearly show that the percentile estimated coverage for the original series is approximately
equal to 0.95, which is very close to the theoretical range of the significance level, while the
coverage of the percentile *t*-method interval is a little small, which prove that this sampling
method can be used to statistically and reliably infer the parameter. Regardless of the level for the
missing data, the estimated results of directly using the series with missing values are located in
the regions of the selected sample, this indicates that for the parameter *H* stable estimation results
can be obtained without any preprocessing for the raw time series. Moreover, it also shows that
both the PLI and PBI series have higher coverage ratios when there are fewer missing values, but
when faced the higher levels, a considerable part of estimates for all interpolated series fall outside
the percentile or percentile-*t* intervals.
Another method for evaluating the efficiency in a hypothesis testing problem is the power





function, the larger the power function is, the more effectively it can distinguish the null
hypothesis from the alternative hypothesis. Since the probabilities of the two types of error are
related to the theoretical distributions of the estimators, so they are not easily available, but the
probabilities of them can be calculated by Monte Carlo simulations and bootstrap resampling
methods. In general, the power function is continuous, which reflects the distributions of the two
types of error for an estimator, and in order to easily manipulate in the study, here the discrete
values are used for the investigation. In the following the calculation of the probability of Type Ⅱ
Error is used as an example to illustrate the efficiency of the parameter estimates for the various
time series, and prior to the comparisons the Monte Carlo simulations are used to generate a series
of sample data with $H$ values ranging within 0.1-0.9 with a step 0.1, where the size of each sample
is set to 1000. **Figure 8(a)** and **8(b)** shows the probabilities of the estimates of different
populations falling within the percentile and percentile-$t$ estimated intervals of the incomplete and
various interpolated series with the fixed parameter $H$=0.6, where the curves are averaged from
the five missing levels. Compared with the other series, for the original series the percentile-based
probabilities that the population parameters of $H$=0.5 and 0.7 are the smallest, that is, the
probabilities of making Type Ⅱ Errors for the two neighbor populations are the smallest, which
is obviously more efficient than the direct use of the incomplete data and various kinds of
interpolated data for the hypothesis testing, but there are also some occurrences of errors for the
populations of $H$=0.4 and 0.8. Note that there are the rises of the misjudgment ratios of $\theta$=0.5 for
the processed samples, especially obvious from the percentile-$t$ estimation results. Let's take a
closer look at the internal comparisons of the different degrees of deficiency, for the series with
the missing values, when the alternative hypotheses $\theta$=0.5 or 0.7 are true, as the missing data
increase, the probabilities of making type Ⅱ Errors are gradually increased, even at the fifth
level, the null hypothesis $\theta$=0.6 will not get a correct test via the percentile or percentile-$t$ method,
however, the series processed by the PLI and PBI methods perform much better (**Fig.9**). Moreover,
for all series at the higher missing levels, the probabilities of making Type Ⅱ Errors for the
population parameters $\theta$=0.4 and 0.8 are also increased, meaning that the more the missing data
are, the lower the power test efficiency will be. For the PCSI and PCHI series, they exhibit the
extreme performances, when the missing data are more serious, Type Ⅱ Errors mainly occur at
the populations parameter $H$=0.1, 0.2, and 0.3, so the two interpolation methods should be used





with caution.

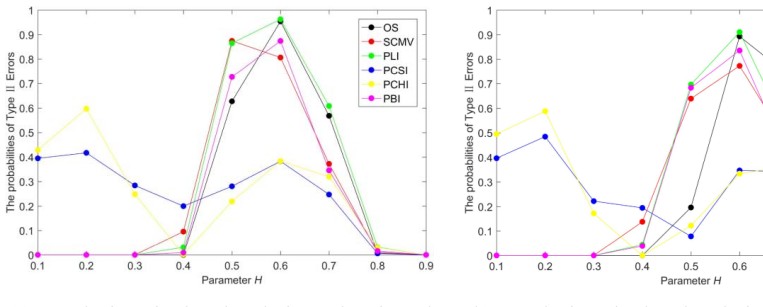

(a) Hypothesis testing based on the intervals estimated          (b) Hypothesis testing based on the intervals estimated
by percentile method                                             by percentile-$t$ method


**Fig.8.** Hypothesis testing of $H$=0.6 against the eight alternative hypotheses with the population parameters ranging
within 0.1-0.9 with a step 0.1 except 0.6, where the size of each sample is set to 1000

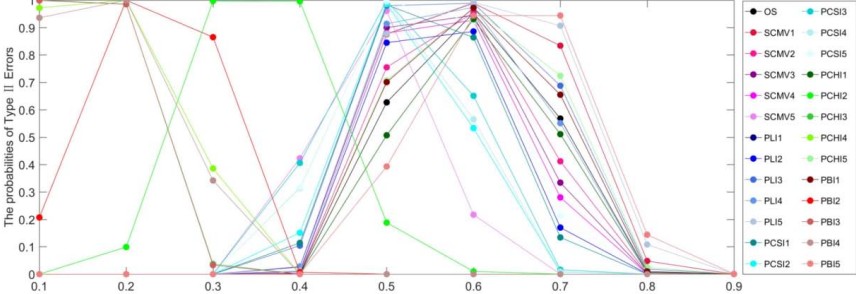


**Fig.9.** Hypothesis testing of $H$=0.6 against the eight alternative hypotheses with the population parameters ranging
within 0.1-0.9 with a step 0.1 except 0.6 at five missing levels.

**3.2 Validation of multifractal data with missing values**

As mentioned above, for the simple case only one parameter $H$ is required for the full

characterization of scaling features for fractional Brownian motion, the estimation results directly
using the incomplete data are less effective than using PLI method for completeness treatment,
and the statistical inference of multifractal parameters will be more complex. From **Table 2** which
shows the RMS errors of the two parameters $C_1$ and $\alpha$, we can see, as a whole, that $C_1$ is less
affected by the interpolation methods, and on the contrary $\alpha$ is affected more, however, $\alpha$ plays an
important role in the identification and judgment of cascade models when multifractal analysis is
used to seek the appropriate model for the natural time series. From the performances of the $\alpha$





estimates, the accuracy using both PLI and PBI methods are better than directly using the
incomplete series, especially for PLI, at the five missing levels, the errors of the two parameters
always oscillate around the values 0.0013 and 0.0831 obtained for the original series, while for the
estimates of $C_1$, the estimation accuracy of the PBI method is higher, however, both PCSI and
PCHI methods exhibit poor performances compared with PLI and PBI methods.

**Table 2** RMS errors of parameters $C_1$ and $\alpha$ calculated for the generalized cascade process containing missing
values (SCMV) and four kinds of interpolated series at five missing levels

| Data | $C_1$ | | | | | $\alpha$ | | | | |
|------|--------|--------|--------|--------|--------|--------|--------|--------|--------|--------|
|      | Level1 | Level2 | Level3 | Level4 | Level5 | Level1 | Level2 | Level3 | Level4 | Level5 |
| SCM  | 0.0026 | 0.0032 | 0.0048 | 0.0055 | 0.0058 | 0.0891 | 0.0999 | 0.1835 | 0.2165 | 0.2262 |
| PLI  | 0.0031 | 0.0038 | 0.004  | 0.0048 | 0.0053 | 0.0806 | 0.0843 | 0.0825 | 0.0844 | 0.0714 |
| PCSI | 0.0043 | 0.0054 | 0.0082 | 0.0106 | 0.012  | 0.3579 | 0.4115 | 0.5413 | 0.6053 | 0.6918 |
| PCHI | 0.0042 | 0.0052 | 0.0072 | 0.0078 | 0.0082 | 0.2485 | 0.3162 | 0.4276 | 0.5125 | 0.5416 |
| PBI  | 0.0013 | 0.0016 | 0.0019 | 0.0026 | 0.0033 | 0.0949 | 0.1118 | 0.1216 | 0.1365 | 0.1619 |


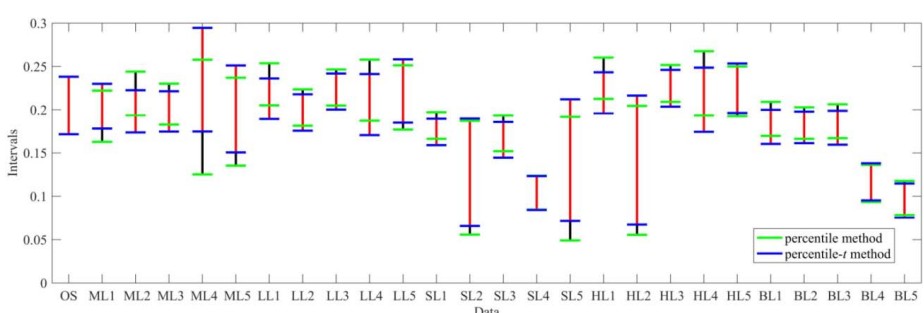


**Fig.10.** The percentile and percentile-$t$ intervals of $C_1$ estimated for the six kinds of time series at five levels based
on RGB resampling mechanism

Similarly, one sample selected from the 1000 Monte Carlo simulations is used to examine the

RGB resampling mechanism for the estimation accuracy of multifractal parameters. **Fig.10** and
**Fig.11** show the percentile and percentile-$t$ estimated intervals of $C_1$ and $\alpha$ for various
experimental data based on the RGB resampling mechanism. It can be seen that for the percentile
and percentile-$t$ estimates, the true value 1.6 of $\alpha$ is not located in the center of the estimated
intervals of the selected series, and all the entire percentile intervals have leftward shifts, while the
shifts for percentile-$t$ method is slightly smaller, but the ranges of the estimated intervals are so
large that this will be bound to influence the power tests. Compared with the estimated intervals of
the selected sample, the left endpoints of the incomplete data for $\alpha$ and $C_1$ are estimated to have

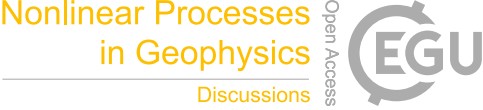



distinct left deviations at the high missing levels, while the PLI method can effectively
compensates for this defect. The figure also shows large differences from the estimated values of
the selected sample for the other three interpolation, except the estimation of the right endpoints
having certain reference values.

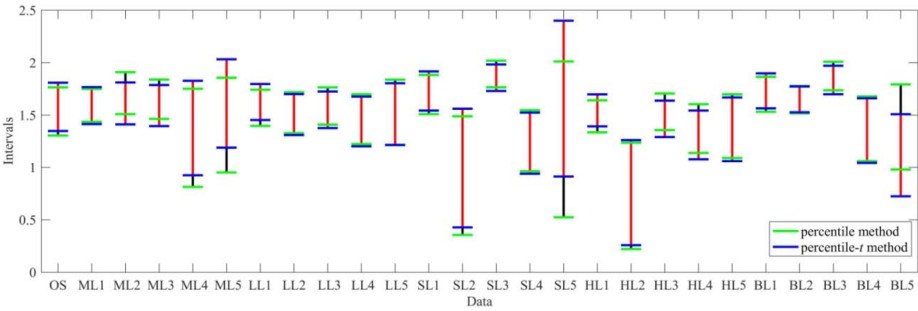


**Fig.11.** The percentile and percentile-$t$ intervals of $\alpha$ estimated for the six kinds of time series at five levels based
on RGB resampling mechanism

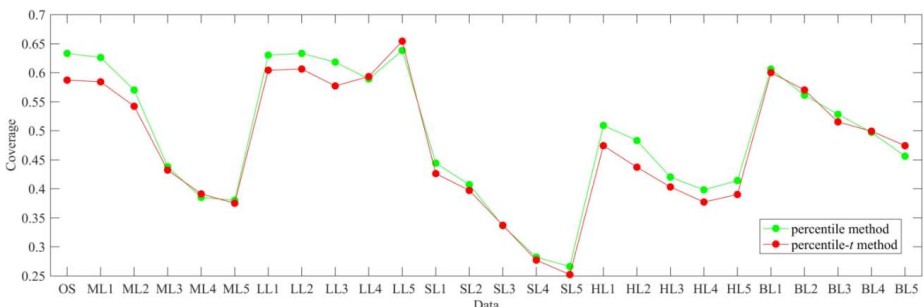


**Fig.12.** The coverage ratios of $\alpha$ estimated for the six kinds of time series at five levels by using percentile and
percentile-$t$ methods

The coverage of the original data, i.e., the percentage of unprocessed Monte Carlo simulation
data falling within the estimated intervals of the selected sample, are approximately 0.71 and 0.63
for $C_1$ and $\alpha$, and there is some deviations from the nominal quantile 0.95, which may be caused
by the sample sizes, but it could constitute the main coverage of Monte Carlo estimations,
therefore there are no fundamental impacts on the estimation accuracies of parameters in the
statistical inference processes and the comparisons among interpolation methods. For the
estimates of $\alpha$ shown in **Fig.12**, the accuracy of PLI method is higher than that of directly using
the incomplete data, and the higher the level of missing, the higher the coverage. For the estimates
of $C_1$ shown in **Fig.13**, the accuracy of the PBI, PLI and incomplete series are not much different,





while the PCHI and PCSI series are inferior to the another two interpolations.

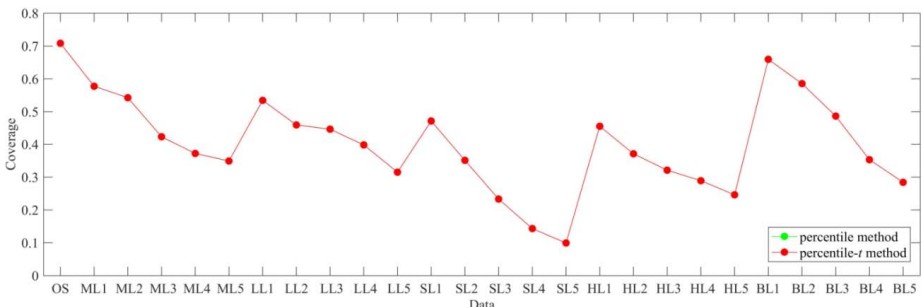

**Fig.13.** The coverage ratios of $C_1$ estimated for the six kinds of time series at five levels by using percentile and
percentile-$t$ methods

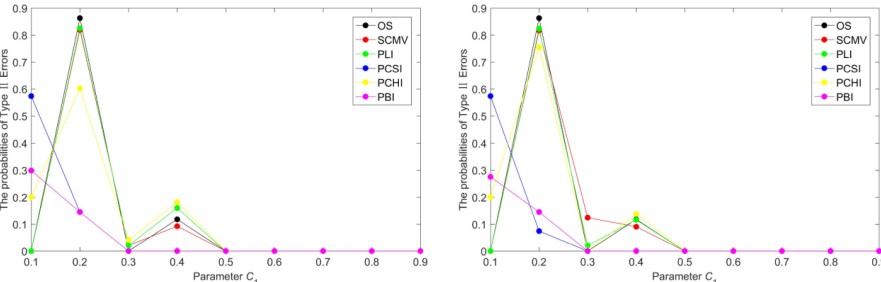

(a) Hypothesis testing based on the intervals estimated        (b) Hypothesis testing based on the intervals estimated
by percentile method                                            by percentile-$t$ method

**Fig.14.** Hypothesis testing of $C_1=0.2$ against the eight alternative hypotheses with the population parameters
ranged within 0.1-0.9 with a step 0.1 except 0.2, where the size of each sample is set to 1000

In order to avoid one of the parameters affecting another in the statistical inference processes,

the power tests of the parameters for cascade series takes a method of maintaining one parameter
unchanged and performing power tests on the other one. Therefore, when keeping $C_1$ unchanged,
$\alpha$ takes the values of 1.1-1.9 with a step 0.1, similarly, and when keeping $\alpha$ unchanged, $C_1$ takes
0.1-0.9 with a step 0.1. Then the probabilities of Type Ⅱ Errors are obtained by the results of the
Monte Carlo estimates for such assigned parameters falling within the confidence intervals of the
selected sample, and **Fig.14** and **Fig.15** show the variations of the occurrence ratios of Type Ⅱ
Errors over different population parameters for the two parameters respectively. For the percentile
results estimated for $C_1$ (**Fig.14(a)**), the probability that the null hypothesis is true is 0.86 when



using the estimated intervals of selected sample, and the probability that $\theta = 0.3$ is misjudged is
less than 0.1. We also get that the test results of original series are significantly better than that of

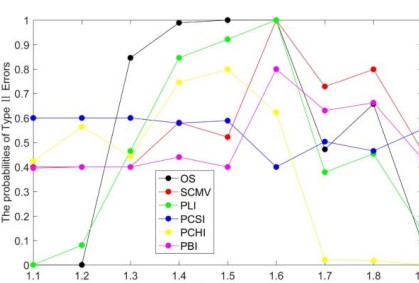

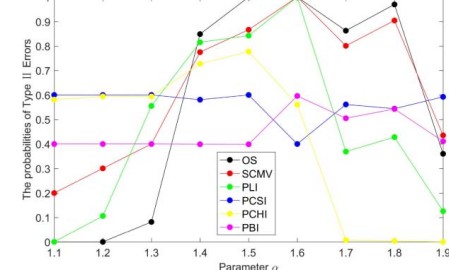

(a) Hypothesis testing based on the intervals estimated
by percentile method

(b) Hypothesis testing based on the intervals estimated
by percentile-*t* method

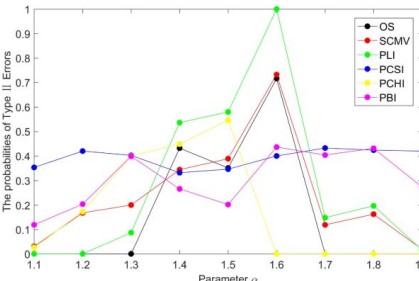

(c) Hypothesis testing based on the intervals estimated
by percentile method at the significance level 0.5


**Fig.15.** Hypothesis testing of $\alpha = 1.6$ against the eight alternative hypotheses with the population parameters ranged
within 1.1-1.9 with a step 0.1 except 1.6, where the size of each sample is set to 1000

the incomplete and interpolated series at any missing levels. On the whole, the incomplete and PLI
series can provide near-original inference results, which outperform the other three interpolation
methods. Among the other three interpolation methods, although some correct judgments can be
made at some levels, the misjudgment probabilities of adjacent parameters cannot be ignored,
even exceeding the probabilities that the null hypotheses are true. Under the test of the
significance level 0.95, the percentile or percentile-*t* estimation accuracy of $\alpha$ for the selected
series is not high, and high error ratios appear on the population parameters ranging from 1.3 to1.9
except 1.6, resulting in the neighboring parameters being misjudged and accepting the null



hypotheses. **Fig.15 (c)** gives the percentile results at the significance level 0.5, and we can get that
the PLI series at high missing levels show greater stability than the incomplete series. Although
the results of the incomplete series at the low missing levels are close to the PLI series, while at
the high missing levels, the probabilities of Type Ⅱ Errors of $\theta$=1.2 and 1.3 are unable to ignore,
just like the estimates of the interval endpoints, the PLI method well avoids the erroneous
judgments. The high error probabilities are still present at $H$=1.3 and 1.4, and the main reasons for
the high error ratios may be the poor stability of the estimation method and that the sample sizes
are too small. Because we focus on the effects on the accuracies caused from the lack of data and
interpolation methods, so these deviations have little effect on the analysis, in addition, you can
also control the level of significance to control the probabilities of falling within the confidence
interval.
**3.3 Validation of actual data with missing values**
In addition to using the simulation data for the examination, this study also uses empirical

data $PM_{2.5}$ series collected in Beijing from January 2016 to December 2016 for one year, with a
total of 8764 data points, and the parameters $C_1$ of $\alpha$ are estimated to be 0.1343 and 1.992
respectively, which indicate the distribution of the series is close to the lognormal distribution. The
examined data are formed according to the preceding procedures in advance, **Table 3** gives the
comparative RMS errors of the two parameters for various time series at all cases. It can be seen
that as the missing level increases, the errors are increased and the accuracy of the estimates
decrease. The accuracy of PLI and PBI methods is slightly better than that of the incomplete data,
while the familiar performances of the estimates as in the simulations examination occur on the
PCSI and PCHI methods.

As can be seen from **Fig.16** and **Fig.17**, the accuracy of the left and right endpoint estimates

of confidence intervals are decreased as the amounts of missing data increase for various series.
For the estimates of $C_1$ (**Fig.16**), the accuracy of PBI and PLI are the highest, and the errors of the
two endpoints for the original series are evidently smaller than the rest of the series, while the
incomplete series will not be correctly estimated. Combined with the performances of the
simulation data, it can be inferred that the parameter $C_1$ is sensitive to the nature of the data,



because $C_1$ reflects the sparsity of the data, therefore, when a data is somewhat or more sparse,
using the incomplete data directly may yield more reasonable result. Compared with the right
endpoint estimates, the accuracy of left endpoint estimates of the two interpolation methods are
higher, except PCSI and PCHI methods. For the estimates of $\alpha$ (**Fig.17**), the incomplete, PLI and
PBI series are all better, and the other two interpolation methods are less accurate. Compared with
the percentile method, the percentile-$t$ interval of the original series is narrowed, and we can get
that the percentile-$t$ method is more accurate by comparing with the endpoints of confidence
intervals estimated for various series.

**Table 3** RMS errors of parameter $C_1$ and $\alpha$ calculated for Beijing PM2.5 time series containing missing values
(SCMV) and four kinds of interpolated series at five missing levels

| Data | $C_1$ | | | | | $\alpha$ | | | | |
|------|--------|--------|--------|--------|--------|--------|--------|--------|--------|--------|
| | Level1 | Level2 | Level3 | Level4 | Level5 | Level1 | Level2 | Level3 | Level4 | Level5 |
| SCM | 0.0006 | 0.0016 | 0.0029 | 0.0044 | 0.0061 | 0.0083 | 0.0096 | 0.0092 | 0.0107 | 0.0134 |
| PLI | 0 | 0.0001 | 0.0001 | 0.0001 | 0.0001 | 0.004 | 0.008 | 0.0116 | 0.0125 | 0.0126 |
| PCSI | 0.0499 | 0.0931 | 0.1342 | 0.1526 | 0.1801 | 0.1049 | 0.1971 | 0.2484 | 0.3004 | 0.2998 |
| PCHI | 0.0191 | 0.0444 | 0.0656 | 0.0882 | 0.1032 | 0.0757 | 0.1305 | 0.1767 | 0.1678 | 0.1894 |
| PBI | 0 | 0 | 0.0001 | 0.0001 | 0.0002 | 0.0027 | 0.0054 | 0.0086 | 0.0116 | 0.0133 |


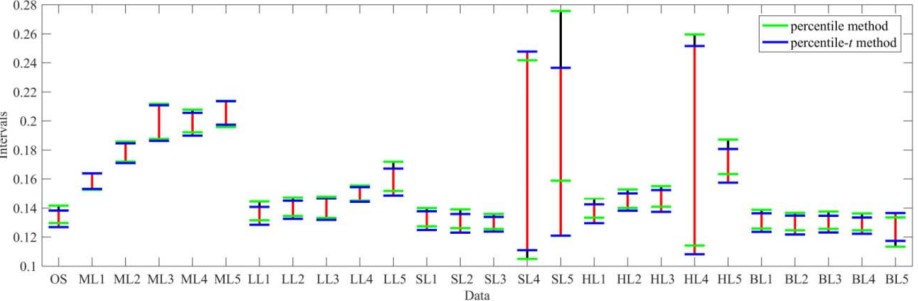


**Fig.16.** The intervals of $C_1$ estimated for the six kinds of time series constructed from Beijing PM2.5 time series at
five levels by using percentile and percentile-$t$ methods

From the coverage results of the confidence intervals estimated by using bootstrap method,

the percentile coverage ratios of PLI at all missing cases exceed 40%, and this indicates that its
accuracy is the highest (**Fig.18** and **Fig.19**). The sensitivity of $C_1$ to the missing values lead to all
the estimates exceeding the interval of the original series from percentile or percentile-$t$ results for
the series with missing values, and for all series, the accuracy of the $\alpha$ estimates is better than that


of $C_1$. For the estimates of $C_1$, as for the slight differences of the coverage between the percentile
method and the percentile-$t$ method, the reason is that the interval estimates using percentile-$t$
method of the original series is narrowed. Since there are no repeated observations for a fixed
parameter, so the empirical series is no longer to perform the power test here.

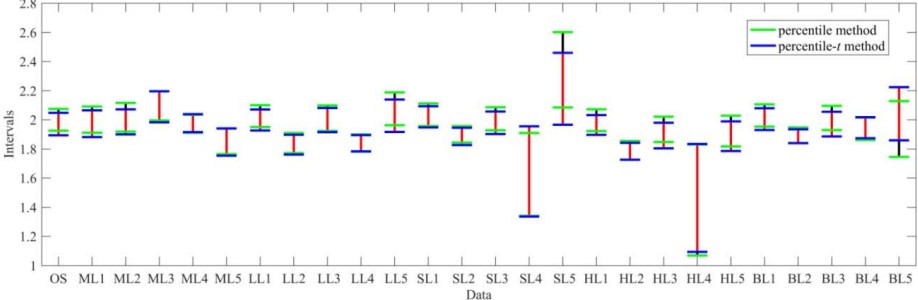

**Fig.17.** The intervals of $\alpha$ estimated for the six kinds of time series constructed from Beijing PM2.5 time series at
five levels by using percentile and percentile-$t$ methods

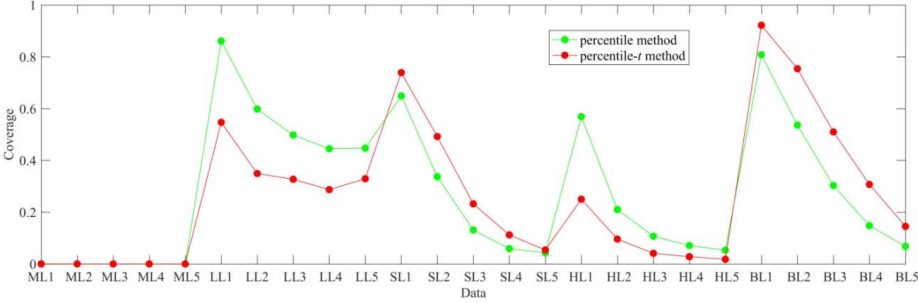

**Fig.18** The coverage ratios of $C_1$ estimated for the six kinds of time series constructed from Beijing PM$_{2.5}$ at five
levels by using percentile and percentile-$t$ methods

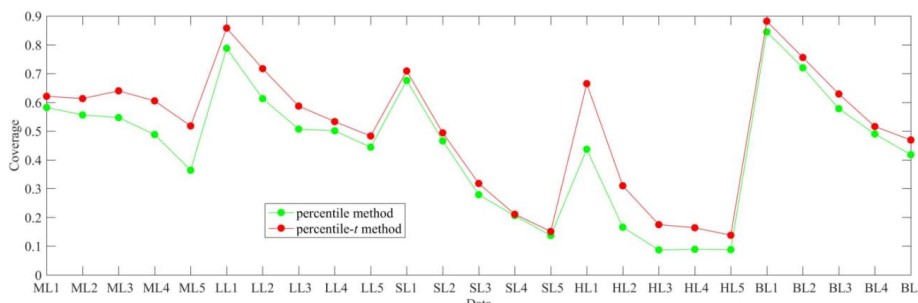

**Fig.19.** The coverage ratios of $\alpha$ estimated for the six of time series constructed from Beijing PM$_{2.5}$ at five levels
by using percentile and percentile-$t$ methods
**4 Conclusions**

An observation series obtained in the natural world is often incomplete and contains many



missing values, therefore, the fractal modeling of a time series with missing values has certain
uncertainties, or the problem is whether it is essential to perfect the data with missing values prior
to the multifractal analysis. As we all know, a fractal series is irregular and scalar, but these
interpolation methods do not consider the scaling characteristics of the series, such as the four
interpolation methods adopted in the paper, while they make the data complete by using the
relationships between the missing points and its adjacent sample values, therefore these
interpolation methods can not completely replace the losses caused by missing data, and then how
about the accuracy of the estimates of parameters for multifractal data, or the maintenance of the
multifractal features for various interpolation methods?
From the results of the RMS errors, PLI method is reliable, which can provide reasonable
estimation accuracy not only for the simulation data such as monofractal or multifractal data, but
also for the experiment data, and the results of the direct use of the incomplete data are proved to
be effective at the low missing levels. Both PCSI and PCHI methods behave badly, and it is not
difficult to find that the result is caused by the smoothness yielded from the interpolation, it is also
concluded that with the increase of the missing values, the influence of PCSI and PCHI methods
on the estimation accuracy gradually become greater than the other three methods.
In order to study the distributions of the estimates of the parameters, a new resampling
method, which focused on fully using the information of the series and is only controlled by the
random numbers, is used to estimate the confidence intervals of the series containing missing
values and various interpolated series. The resampling method was proved effective based on the
fact that the bootstrap intervals with quantile 0.95 can well cover the estimates of the Monte Carlo
simulations. For the estimates of Hurst index $h$, it can be concluded that when the amounts of
missing data are large, the direct use of the series containing missing values cannot be correctly
estimated, but the more accurate estimates can be obtained through PLI and PBI preprocessing.
With some differences between percentile method and percentile-$t$ method, the intervals estimated
for the former are shifted to the left compared to the latter. For $C_1$ and $\alpha$, compared to the
confidence intervals of the original data, the left endpoints of incomplete data are estimated to
have a significant left deviation at the high missing levels, and the right endpoint errors are smaller.
The left endpoint estimates of PLI are close to the original series, and the accuracy improved
significantly compared with the incomplete data. Through the investigation of the experimental





data, it is found that the estimates of $C_1$ are more sensitive to the properties of the data, and it can
not be accurately estimated by directly using the series containing missing values. For the
estimates of $\alpha$, we can see that the incomplete, PLI and PBI series have better performances.
Compared with the percentile method, the percentile-$t$ intervals of the original series are narrowed,
and the accuracy of the percentile-$t$ method is higher by comparing the endpoint estimates of
various series.
The fBm and cascade simulations with known values for parameters were used to study the
probabilities of Type Ⅱ errors, i.e., the probabilities of falling within the confidence intervals of
the selected sample estimated by bootstrap method for such fixed population parameters. For fBm,
eight alternative hypotheses were assigned, such as 0.1, 0.2, . . . 0.9, except 0.6, while for cascade,
keep one parameter unchanged, and let another take a value in a range 1.1-1.9 or 0.1-0.9
respectively. For Hurst index $h$, it was analyzed that the probability of Type Ⅱ error for each
alternative hypothesis is the closest to the original series for directly using the series containing
missing values and PLI method at the low missing levels, which is more effective than the other
three interpolation methods, while at the high missing levels the error ratios of the population
parameters around 0.6 keep rising. For the multifractal series, the similar conclusion can be drawn,
but when faced with low missing levels, the performance of the PLI is even better than using the
series containing missing values directly. Moreover, when the significance level is set to 0.5, it can
be concluded that the PLI method has more stability than directly using the series containing
missing values.

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
