# Peer review of "Exploring the effects of missing data on the estimation of fractal and multifractal parameters based on bootstrap method"

_Nonlinear Processes in Geophysics, 2018_

## Referee Comment (RC1) · Anonymous Referee #1 · 15 Dec 2018

The writing of this manuscript is poor and the English language needs revision. I do not recommend to accept this manuscript. It needs complete rewriting.

Form the beginning this manuscript is poorly written. List of problems in the introduction: Line 77: what id PM2.5? Line 78: what is DEM? Line 80: how is H defined? The H value in itself does not prove that the studied process is a Brownian motion or a fractional Brownian motion. Line 82: what is alfa? Line 105: what is the relation of this scientific ms. with comparative politics or international relations? Line 194: there is a - sign in the exponent

I could follow (with pain) the idea of the manuscript until section 2.3. I could not under-

stand this section. Why is resampling needed?

Globally the problem of missing values in measured time series is very relevant and the question of parameter estimation for scaling processes is a legitimate question. But this manuscript suffers many weaknesses: - there is no review of the literature on this topic. Many papers have been devoted to such general topic. The authors should also mention the estimation of spectral slopes of scaling precesses with missing or irregular data, for which there is a vast literature; - the authors should quantify and characterize the missing values: irregular missing values, or by blacks, and how many compared to the total length of the series? - why perform classical interpolations for a scaling fractal or multifractal field having huge fluctuations? - the authors should separate pure cascade processes from non stationary processes with stationary increments (fBm) since the results will be clearly different; - why is bootstrap necessary for such topic? - for numerical studies, very long time series with many realizations should be considered (with billions of data points) and not 1000 or 2048 data points (lines 350-354).

---

## Referee Comment (RC2) · Anonymous Referee #2 · 23 Dec 2018

The article treats a very interesting subject and distinguishes itself by comparing a wide range of interpolation methods. The subject is vast and an exhaustive treatment is not trivial and I commend the authors for their work.

On the face of the results, it appears to me none of the interpolation methods clearly outperform the basic method of using "series containing missing values". Given that the method, which appears to be the same as detrended fluctuation analysis (DFA), can accommodate missing data, I would see the pure randomly gliding boxes method as the clear winner over the interpolation methods. This is interesting as it confirms the intuition that when interpolation can be avoided, it should. I think such studies would

be more interesting to perform with methods such periodogram or multi-tapers which usually require uniform data and thus interpolation, rather than with DFA which does not require interpolation.

However, I cannot recommend acceptance of the paper at the moment for reasons of language, order and content. At best, there should need to be major revisions made in both the writing and the numerical experiments, but given the large amount of changes I believe it requires, I think that it will be easier for the authors to resubmit the paper anew. Below I give evidence of the need for major revisions.

Language

I started making detailed comments on the improvements needed in the writing of sentences, but then stopped writing them all since most sentences need to be revised. Here is the beginning:

Line 53: Define the following acronyms: PLI and PBI

Line 55: Replace 'are unable to ignore' by 'cannot be ignored'

Line 82: I think you did not mean to have '=' there in '0<alpha=<1'

Line 164: I fail to see a difference between regression residual variance and detrended fluctuations analysis (DFA). Did you mean DFA?

Line 163-176: It seems to me that the explanation of the regression residual variance is misplaced, right in the middle of the explanation about fractional Brownian motion.

Line 195: "the" scaling function (an article is needed in front of scaling)

Line 197: "a" universal generalized (u is pronounced like a consonant here, therefore "a" rather than "an".)

Line 197: Schertzer and Lovejoy missing from reference list at the end.

Line 200: Replace "of which have" by "used are"

Line 201: Replace "involving lots" by "it involves a lot"

Line 242-245: Sentence is convoluted, needs rewrite.

Line 246-248: Sentence is convoluted, needs rewrite.

Line 247: On the condition that it is poorly known? I think you mean "even when it is poorly known".

Line 255-258: Sentence is convoluted, needs rewrite.

Line 260: "By the glide of a group of boxes"? I think "using a group of gliding boxes" is what you mean.

Line 262-264: Sentence is convoluted, needs rewrite.

Order

Then, another part of the writing is what I will call here order, and in which I mostly include problems of definitions. The symbols used are rarely defined properly (F, G, Theta, sigma, s, c, etc). This ties in with problems of structure. The biggest problem I identified is the lack of numbered equation. Any equation which is important warrants a number. I think a good way to go about writing the theory section, is to introduce first an equation, and then define in the most straightforward manner, what the terms in the equations stand for, and then give explanations of the significance of the terms and the equation. Below, I give a few points as proof again that major revisions are needed in this aspect.

Line 204: I don't think int() is a standard function, you should indicate that it means you are taking the integer part of the fraction.

Line 278: What is meant by concealment?

Line 278: If there is such "concealment" to produce Q, then Q isn't really a population anymore, but rather a sample right?

Line 327: What is meant by standard deviation of distribution with double sampling?

Line 324, 328: R and S are the re-sampling and secondary sampling, but I do not think these were defined clearly. It would help to indicate on the diagrams when the sampling R and S happen. On this note, I am unsure of the reason for the necessity of doing this resampling twice. I think it would be good to explain this better.

Fig 3, 4: Captions should indicate directly which is (a) and which is (b) without the need to refer to the text.

Fig 14: It does not make sense to test for C1=0.9 here. The range of tests could be restricted under 0.5, and the resolution increased to maybe 0.05. Given the unsmoothness and lack of monotonicity, it appears that the number of replicas in the sample might not be sufficient, or at least, I do not see how it makes sense that there is a bump at 0.4.

Line 392: Why is it decreasing? Is that a feature one should expect? Could it be random because the variance of the estimator becomes increasingly large?

Line 427: What is this "power function". I do not think it was properly introduced, and the usage made of it remains vague.

All tables: The numerical results should be given with consistent significant figures

Line 569: Again, giving 1.992 seems overly precise. Generally, the number of sig figs corresponds to the presumed accuracy of the estimates. Unless the accuracy can be to three significant figures, which is not the case here since the error seems to be on the order of 0.1, then it should be rounded to 1 sig figs, i.e. this result could be reported as 2.0 +/- 0.1 and the mean estimate would be virtually equal to a log-normal distribution.

Content

The numerical experiments could be greatly expanded. The most ominous point is that only one set of parameters if tested for each type of series, i.e. fBm and multifractals.

The authors should be aware that the results and conclusions they make, might very well be dependant on the value of those parameters, and I expect they are. So while one type of interpolation method might perform well for H=0.6, another might perform better for H=0.3, and so on. Same applies for C1 and alpha. Therefore, for the paper to be more relevant, I would recommend expanding the experiment to all values of H between 0.1 and 0.9, and all values of C1 between 0 and 0.3, and all values of alpha between 1. and 2. Also, in addition to the RMS value, I think the authors should consider reporting the bias and variance of the estimators. Of course, the authors will realize when they do that, that the vastness of results produced is challenging to report, and that new methods of visualization might be necessary. I believe this will also greatly affect their outlook on their conclusion and the significance, and what is important to report.

---

## Author Comment (AC1) · 29 Jan 2019

**Dear Editors and Reviewers:**

Thank you for your letter and for the reviewers' comments concerning our manuscript entitled "Exploring the effects of missing data on the estimation of fractal and multifractal parameters" (ID: npg-2018-38). Those comments are all valuable and very helpful for revising and improving our paper, as well as the important guiding significance to our researches.

**Anonymous Referee #1:**

Firstly, we sincerely apologize for our poor English writing in our submitted manuscript.

As for the scientific issues discussed in this article, we mainly focused on the impacts of missing data on the estimation accuracy for fractal or multifractal series. To solve this problem, we think there should be many perspectives. It is an angle for stochastic process theory to solve this kind of problem. We think that non-parametric statistical inference based on simulated data is also a relatively simple method. The reason is that statistical inference theory can compare and pick out the advantages or disadvantages of multiple test schemes. As for why we choose various interpolation methods as candidates, such consideration arose from many previous articles using interpolation methods to preprocess them.

We recognized that this paper treated missing data roughly, and does not deeply considered the missing types such as stationary vs. nonstationary, irregular or blank, and so on. Many traces of artificial manipulations are found in the experimental scheme, indeed in this way the missing extent is also emphasized and is supposed to have most influence for the estimations. We think that this method can basically reflect the comparison of the estimation accuracy between the series with missing data and various interpolation methods. Of course, it can also determine whether the interpolation pretreatment is necessary or not.

We try to continue to reinforce some of the conclusions in this article. For instance, the simulation data of fractal Brownian motion with parameter H ranging between 0.1 and 0.9 and multifractal series with parameter $C_1$ ranging between 0.1 and 0.3 and α between 1.0 and 2.0 respectively are generated, and the amount of simulated data is many times larger than before. Monte Carlo and bootstrap statistical inference are used for multiple sample data and single sample data, respectively. Each validation will cover three parts: firstly, the median values are compared for all datasets reflect the degrees of deviation from the true values (**Fig.1 and Fig.2**); secondly, the percentage of estimates to the data with missing values and interpolated data falling into the estimated intervals of the original data are compared to reflect the discreteness of the estimates for the affected data (**Fig.3**); thirdly, the probabilities that the affected population are misjudged as another population, i.e., type I errors are calculated and used to compare the estimation accuracy (**Fig.4**).

Note:

1. 3 missing types, (The missing data are divided into three levels according to the missing degree, and it reflects the size of the missing quantity. The smaller the number, the smaller the degree of missing data.) ;

2. 5 interpolation methods include piecewise linear interpolation (PLI), Nearest-neighbor interpolation (NI), piecewise cubic spline interpolation (PCSI), piecewise cubic Hermite interpolation (PCHI) and piecewise Bessel interpolation (PBI)).

**Anonymous Referee #2:**

Firstly, we sincerely apologize to the reviewers for the poor quality of English writing. Thank you very much for giving this major review opportunity.

Since receiving the feedback, we have devoted ourselves to the revision of the manuscript. We deeply rethink the scientific issues discussed in this paper. Our thinking has changed from emphasizing the using of bootstrap method to the influence of missing data on estimation accuracy for fractal or multifractal series. Meanwhile,we also expand the breadth and depth of the experiment in the paper according to the comments. The amount of simulated data is many times larger than before, which makes it impossible to complete the revision within given time limit. Therefore, we hope to give us another three months to complete the revision.

In response to the reviewer's comments, the simulation data of fractal Brownian motion with parameter H ranging between 0.1 and 0.9 and multifractal series with parameter $C_1$ ranging between 0.1 and 0.3 and α between 1.0 and 2.0 respectively are generated. Monte Carlo and bootstrap statistical inference are used for multiple sample data and single sample data, respectively. The experimental data set include the original data (Monte Carlo simulations), the series with missing data (The missing data are divided into three levels according to the missing degree, and it reflects the size of the missing quantity. The smaller the number, the smaller the degree of missing data.) and interpolated data (For 3 missing types, the interpolated data are obtained using 5 interpolation methods including piecewise linear interpolation (PLI), Nearest-neighbor interpolation (NI), piecewise cubic spline interpolation (PCSI), piecewise cubic Hermite interpolation (PCHI) and piecewise Bessel interpolation (PBI)). Each validation will cover three parts: firstly, the median values are compared for all datasets reflect the degrees of deviation from the true values (**Fig.1 and Fig.2**); secondly, the percentage of estimates to the data with missing values and interpolated data falling into the estimated intervals of the original data are compared to reflect the discreteness of the estimates for the affected data (**Fig.3**); thirdly, the probabilities that the affected population are misjudged as another population, i.e., type I errors are calculated and used to compare the estimation accuracy (**Fig.4**).

[Figure]

[Figure]

Median estimates of α when C₁=0.1, 0.2 and 0.3

Median estimates of C₁ when α ranges between 1.0 and 2.0

**Fig.1**

[Figure]

Median estimates of α for 3 missing types when C₁=0.1, 0.2 and 0.3

[Figure]

Median estimates of α for 5 interpolated series to 3 missing types when C₁=0.1

[Figure]

Median estimates of α for 5 interpolated series to 3 missing types when C₁=0.2

[Figure]

Median estimates of α for 5 interpolated series to 3 missing types when C₁=0.3

**Fig.2**

[Figure]

Percentage of population estimates for missing types falling into the Monte Carlo intervals

[Figure]

Percentage of population estimates for interpolated series falling into the Monte Carlo intervals

**Fig.3**

[Figure]

[Figure]

**Type I errors of *α*=1.5 for 3 missing types**  **Type I errors of *α*=1.5 and C₁=0.1 for 5 interpolated series from 3 missing types**

**Fig.4**